# [Re] Projection-based Algorithm for Updating the Truncated SVD of Evolving Matrices

## Reproducibility Summary

## Scope of Reproducibility

Kalantzis et al. [11] present a method to update the rank-$k$ truncated SVD of matrices where the matrices are subject to periodic additions of rows or columns. The main claim of the original paper states that the presented algorithms outperform other state-of-the-art approaches in terms of accuracy and speed. However, no results were given comparing the proposed methods to other state-of-the-art methods. Accordingly, we reproduce their results and compare it to the state-of-the-art `FrequentDirections` streaming algorithm [6].

## Methodology

We re-implemented the algorithm in Python and evaluated the performance on five datasets. All experiments were run on a MacBook Pro and the code is available on GitHub[1]. The accuracy of the methods were evaluated using the same metrics as in the paper.

## Results

We successfuly reproduced the task-agnostic experiments of the original paper, finding our results to strongly match with the original results. We also carried out a comparison with `FrequentDirections` but found the evaluation metrics of the original paper to be ill-suited to compare - setting up for further work on developing fair comparisons.

## What was easy

The benchmark algorithm was fairly simple to implement. Furthermore, running the experiments did not place any computational resource burden as all experiments could be run on a laptop.

## What was difficult

The most difficult part of the reproduction study was understanding the justification underlying the construction of the algorithm as it involved several complex proofs from numerical linear algebra to provide bounds on the accuracy. Demystifying the specifics of constructing the projection matrix for the main algorithm the author's propose was also initially difficult until we gained access to their code.

## Communication with original authors

We contacted one of the authors by email and received their data and MATLAB implementation of the algorithm and experiments.

---

[1]https://anonymous.4open.science/r/truncatedSVD-0162/

Submitted to ML Reproducibility Challenge Fall 2021. Do not distribute.

# 1 Introduction

The singular value decomposition (SVD) remains a fundamental dimensionality reduction technique in machine learning and continues to be used in a variety of applications. In a traditional formulation, the entirety of the matrix to be decomposed is available at the time of application of the SVD. However, certain applications, such as latent semantic indexing (LSI) and recommender systems, have matrices that are subject to the periodic addition of new rows and/or columns. A naïve solution is to recalculate the SVD each time the matrix is updated, but such an approach quickly becomes impractical when updates are frequent. For this reason, algorithms that exploit information on the previous SVD of the matrix to calculate the SVD of the updated matrix are crucial. Such schemes have been proposed for both the full SVD and rank-$k$ SVD. The algorithm presented in [11], which is the focus of our study, is for the rank-$k$ truncated SVD case.

Following the notation introduced in [11], the problem of updating the rank-$k$ truncated SVD of an updated matrix is as follows. Let $B \in \mathbb{C}^{m \times n}$ be a matrix for which a rank-$k$ SVD $B_k = U_k \Sigma_k V_k^H = \sum_{j=1}^{k} \sigma_j u^{(j)} (v^{(j)})^H$ where $U_k = [u^{(1)}, \ldots, u^{(k)}]$, $V_k = [v^{(1)}, \ldots, v^{(k)}]$, and $\Sigma_k = \texttt{diag}(\sigma_1, \ldots, \sigma_k)$ where $\sigma_1 \geq \sigma_2 \geq \cdots \geq \sigma_k > 0$ is known. The goal is to approximate the rank-$k$ SVD $A_k = \widehat{U}_k \widehat{\Sigma}_k \widehat{V}_k^H = \sum_{j=1}^{k} \widehat{\sigma}_j \widehat{u}^{(j)} (\widehat{v}^{(j)})^H$ of the updated matrix

$$A = \begin{pmatrix} B \\ E \end{pmatrix}, \text{ or } A = \begin{pmatrix} B & E \end{pmatrix}$$

where $E \in \mathbb{C}^{s \times n}$ or $E \in \mathbb{C}^{m \times s}$ is the matrix containing the newly added rows or columns, respectively. We focus on the row-update case in this study as is the case in [11].

The remainder of this study is outlined as follows. In Section 2, we introduce the central claim of the original paper that we tested in our study. Following that, in Section 3, we introduce the necessary background prior to describing the proposed algorithm. In Section 4, we describe the experimental setup: our implementation of the algorithm, datasets used, and experiments run. We present the experimental results in Section 5 along with our interpretation of the results and thoughts on the overall study in Section 6.

# 2 Scope of reproducibility

In this study, we aimed to verify the central claim of the original paper, which stated that the proposed algorithm outperforms other state-of-the-art approaches at calculating the truncated SVD of evolving matrices. In particular, they claimed that the method had especially high accuracy for the singular triplets with the largest modulus singular values. We sought to verify this claim by evaluating two metrics using our implementation of the method as well as with `FrequentDirections`, a state-of-the-art matrix sketching and streaming algorithm [6]:

1. Relative approximation error `rel_err` of leading $k$ singular values of $A$ (Equation 1) is smaller when using the proposed algorithm compared to previous methods.

$$\texttt{rel\_err} = \left| \frac{\widehat{\sigma}_i - \sigma_i}{\sigma_i} \right| \tag{1}$$

2. Scaled residual norm `res_norm` of leading $k$ singular triplets $\{\widehat{u}^{(i)}, \widehat{v}^{(i)}, \widehat{\sigma}_i\}$ (Equation 2) is smaller when using the proposed algorithm compared to previous methods.

$$\texttt{res\_norm} = \frac{\left\| A \widehat{v}^{(i)} - \widehat{\sigma}_i \widehat{u}^{(i)} \right\|_2}{\widehat{\sigma}_i} \tag{2}$$

Additionally, we also sought to verify the original paper's claims about the runtime performance of the proposed algorithm.

# 3 Projection-based update algorithm

In the following sections, we first introduce the original `zha-simon` algorithm, then introduce the proposed projection-based update algorithm. Note that there are two implementations to the proposed algorithm: one which uses the same projection matrix as the `zha-simon` algorithm (Algorithm 2.1) and another that uses an enhanced projection matrix (Algorithm 2.2).

## 3.1 Zha-Simon algorithm

As motivated in the introduction, an update algorithm that uses prior knowledge regarding the SVD of the matrix is crucial for it to be useful in practice. The algorithm proposed in [11] is based on an algorithm proposed in [15], the latter of which we will refer to as the `zha-simon` algorithm (Algorithm 1). Using `zha-simon` in the row-update case $A = \begin{pmatrix} B \\ E \end{pmatrix}$, the QR decomposition of the row space of $E$ that is not captured by the range of the right singular vectors $V_k$ can be expressed as $(I - V_k V_k^H)E^H = QR$. Using this result and the previously known rank-$k$ SVD $B_k = U_k \Sigma_k V_k^H$, the updated matrix $A$ can be decomposed approximately as follows:

$$A = \begin{pmatrix} B \\ E \end{pmatrix} \approx \begin{pmatrix} U_k \Sigma_k V_k^H \\ E \end{pmatrix} = \begin{pmatrix} U_k & \\ & I_s \end{pmatrix} \begin{pmatrix} \Sigma_k & \\ EV_k & R^H \end{pmatrix} \begin{pmatrix} V_k^H \\ Q^H \end{pmatrix} \tag{3}$$

If we let $F\Theta G^H$ be the compact SVD of $\begin{pmatrix} \Sigma_k & \\ EV_k & R^H \end{pmatrix}$, then Equation 3 can be further decomposed as follows:

$$A \approx \begin{pmatrix} U_k & \\ & I_s \end{pmatrix} (F\Theta G^H) \begin{pmatrix} V_k^H \\ Q^H \end{pmatrix} = \left( \begin{pmatrix} U_k & \\ & I_s \end{pmatrix} F \right) \Theta \left( \begin{pmatrix} V_k & Q \end{pmatrix} G \right)^H \tag{4}$$

The key here is to notice that the approximation of the rank-$k$ truncated SVD of $A$ using the `zha-simon` algorithm does not require access to the previous matrix $B$ – only the rank-$k$ SVD $B_k = U_k \Sigma_k V_k^H$ of the matrix from the previous iteration is needed. We can further simplify Equation 4 and see that it approximates the SVD of $A$ as $A \approx (ZF)\Theta(WG)^H$ where $Z = \begin{pmatrix} U_k & \\ & I_s \end{pmatrix}$ and $W^H = \begin{pmatrix} V_k & Q \end{pmatrix}^H$ are orthonormal matrices with ranges that approximately capture $\mathtt{range}(\widehat{U}_k)$ and $\mathtt{range}(\widehat{V}_k^H)$, respectively.

---

**Algorithm 1** `zha-simon` algorithm

**Input:** $A, E, U_k, \Sigma_k, V_k, k$
1: $Z \leftarrow \begin{pmatrix} U_k & \\ & I_s \end{pmatrix}$
2: $[Q, R] \leftarrow \mathtt{qr}(I - V_k V_k^H)E^H$
3: $W \leftarrow \begin{pmatrix} V_k & Q \end{pmatrix}$
4: $[F_k, \Theta_k, G_k] \leftarrow \mathtt{svd}(Z^H AW, k)$
5: $\overline{U}_k \leftarrow ZF_k$
6: $\overline{\Sigma}_k \leftarrow \Theta_k$
7: $\overline{V}_k \leftarrow WG_k$
**Output:** $\overline{U}_k \approx \widehat{U}_k, \overline{\Sigma}_k \approx \widehat{\Sigma}_k, \overline{V}_k \approx \widehat{V}_k$

---

**Algorithm 2** Proposed row-update algorithm

**Input:** $B, E, k$
1: $[U_k, \Sigma_k, V_k] \leftarrow \mathtt{svd}(B, k)$
2: Construct projection matrix $Z$
3: $[F_k, \Theta_k] \leftarrow \mathtt{svd}(Z^H A, k)$ where $A = \begin{pmatrix} B \\ E \end{pmatrix}$
4: $\overline{U}_k \leftarrow ZF_k$
5: $\overline{\Sigma}_k \leftarrow \Theta_k$
6: $\overline{V}_k \leftarrow A^H \overline{U}_k \overline{\Sigma}_k^{-1}$
**Output:** $\overline{U}_k \approx \widehat{U}_k, \overline{\Sigma}_k \approx \widehat{\Sigma}_k, \overline{V}_k \approx \widehat{V}_k$

---

### 3.2 Proposed row-update algorithm

In practice, computing the rank-$k$ truncated SVD of $A$ using Algorithm 1 is expensive due to the QR (Step 2) and SVD (Step 4) steps and possibly inaccurate based on the structure of $A$ [11]. The cost of the QR decomposition can be mitigated by setting $W = I_n$ by observing that $\widehat{v}^{(i)} \subseteq \mathtt{range}(I_n)$ for $i = 1, \ldots, n$. Therefore, $Z^H AW$ in Step 4 can be replaced with $Z^H A$ and the QR decomposition in Step 2 can be eliminated. With these modifications, we have the new proposed row-update algorithm (Algorithm 2). Note that Step 2 has intentionally not been specified as the authors proposed two options for the construction of the projection matrix $Z$.

The first option (Algorithm 2.1) uses the same $Z$ matrix as in Algorithm. Although the construction of $Z$ and $Z^H A$ are presented in two separate steps in Algorithm 2, $Z^H A$ for Step 3 is directly computed as 1. Below are the expressions for $Z$ and $Z^H A$ for Algorithm 2.1.

$$Z = \begin{pmatrix} U_k & \\ & I_s \end{pmatrix} \tag{5a}$$

$$Z^H A = \begin{pmatrix} \Sigma_k V_k^H \\ E \end{pmatrix} \tag{5b}$$

In the case where the rank of $B$ is larger than $k$ and the singular values $\sigma_{k+1}, \ldots, \sigma_{\min(m,n)}$ are not small, the approximation returned by Algorithm 2.1 can be of poor accuracy. Algorithm 2.2 addresses this by using an enhanced version of the projection matrix by adding a term $-B(\lambda)BE^H$ in the $Z$ matrix such that

$$Z = \begin{pmatrix} U_k & -B(\lambda)BE^H \\ & & I_s \end{pmatrix} \tag{6}$$

Setting $X = -B(\lambda)BE^H$, the additional term is equal to the matrix $X$ that satisfies the equation

$$-(BB^H - \lambda I_m)X = (I_m - U_k U_k^H)BE^H, \tag{7}$$

which can be computed using the block conjugate gradient (BCG) method [12]. To ensure that the matrix $-(BB^H - \lambda I_m)$ is positive definite for BCG, a lower bound of $\lambda > \sigma_1^2$ is imposed. The leading singular value can be estimated using a few iterations of truncated SVD. However, to reduce the number of columns in $X$ and keep $Z$ manageable, the randomized rank-$r$ SVD of $X$ can be taken so that

$$-B(\lambda)BE^H R \approx X_{\lambda,r} S_{\lambda,r} Y_{\lambda,r}^H \tag{8}$$

where $R$ is a matrix with at least $r$ columns whose entries are i.i.d. Gaussian random variables with zero mean and unit variance. With $X_{\lambda,r}$, the $Z$ and $Z^H A$ matrices can be calculated as

$$Z = \begin{pmatrix} U_k & X_{\lambda,r} \\ & & I_s \end{pmatrix} \tag{9a}$$

$$Z^H A = \begin{pmatrix} \Sigma_k V_k^H \\ X_{\lambda,r}^H B \\ E \end{pmatrix} \tag{9b}$$

For more detailed explanations and derivations of the algorithms and their associated proofs, we refer readers to [11].

# 4 Methodology

Professor Vassilis Kalantzis, who we contacted via email, generously provided us with the relevant MATLAB code and data; however, we chose to re-implement the algorithm from scratch in Python with standard packages (NumPy [9], SciPy [14], and scikit-learn [13]) and used the MATLAB code to confirm our implementation. We compared the performance of Algorithms 2.1 and 2.2 with `FrequentDirections` [6], a state-of-the-art streaming algorithm. Experiments were conducted on a MacBook Pro with a 2.3 GHz Dual-Core Intel Core i5 processor with 16 GB of RAM, and the code is publicly available on GitHub[2]. All plots were generated using Matplotlib [10].

## 4.1 Implementation

We chose to implement the three truncated SVD update algorithms as methods of an `EvolvingMatrix` class, which we will refer to as `EM` from here on out. With each experiment, the `EM` class was initialized with various parameters (initial matrix, matrix to be appended, number of batches, etc.) and updates were carried out using one of the update methods. A simplified version of the experiment is shown in Listing 1. Algorithms 2.1 and 2.2 were written based on the pseudo-code presented in Algorithm 2, where the $Z$ and $Z^H A$ matrices were calculated using their respective formulas.

**Algorithm 2.1** The $Z$ and $Z^H A$ matrices were constructed as in Equations 5a and 5b, respectively.

**Algorithm 2.2** The main difficulty in implementing Algorithm 2.2 was in the calculation of $X_{\lambda,r}$. We chose to solve for $X$ in Equation 7 using the block Conjugate Gradient method (BCG) [12] as recommended in [11]. Though [11] specified, at maximum, one iteration of BCG, we found that the MATLAB code set the limit to two iterations. As the additional iteration did not greatly increase the computational cost, we chose to run BCG a maximum of two iterations as well. Once $X$ was calculated, we calculated $X_{\lambda,r}$ as per Equation 8 using randomized SVD [7]. For this, we used the scikit-learn `randomized_svd` implementation [13]. Based on the description for calculating $X_{\lambda,r}$ in [11], we set `n_components`$= r$, `n_oversamples`$= 2r$, and `n_iter`$= 0$. The $X_{\lambda,r}$ returned was then used to calculate $Z$ and $Z^H A$ as in Equations 9a and 9b, respectively.

---

[2]https://anonymous.4open.science/r/truncatedSVD-0162/

```
1  # Initialize EM object with initial matrix, number of batches, and desired rank
2  model = EM(initial_matrix, n_batches, k_dim)
3
4  # Set entire matrix to be appended
5  model.set_append_matrix(E)
6
7  # Update over specified number of batches
8  for i in range(n_batches):
9      model.evolve()       # append rows to matrix
10     model.update_svd()  # update truncated SVD
11
12     # Calculate metrics for pre-selected updates
13     if model.phi in phis2plot:
14         model.calculate_true_svd()
15         model.save_metrics()
```

Listing 1: Simplified experiment structure

**Frequent Directions**    A modified version of `FrequentDirections`[3] was incorporated as an update method into the EM class. Since `FrequentDirections` is a line-by-line update method as opposed to a batch update method, the update method in the EM class was constructed to receive a matrix $E$ containing the rows to be added and performs the `FrequentDirections` algorithms for each row of the $E$. Any form of error metric calculation or subsequent update is performed only after the entire matrix $E$ has been processed using the line-by-line update method.

Since the updated matrix $B$ for the `FrequentDirections` method has constant dimensions throughout the update process, the residual norm error calculation is modified to measure the error between $B$ and $A'$ where $A'$ is a truncated version of $A$ that only holds the first $2l$ singular vectors and values of $A$ and where $2l$ is the number of rows in $B$.

### 4.2 Datasets

In total, we conducted experiments on five datasets. MED, CRAN, CISI, and Reuters-21578 are term-document matrices from latent semantic indexing applications [1–5] and ML1M is a movie rating dataset from MovieLens [8]. Table 1 lists the dimensions of the matrices as well as the average number of nonzero (nnz) entries per row and Figure 1 shows the leading 100 singular values for each matrix. It should be noted that the matrices used for CISI, CRAN, and MED in [11] had slightly different dimensions compared to what was listed on [1]. We received these datasets along with the MATLAB code and chose to use their versions of the data for ease of comparison; as we were interested in the accuracy of singular value reconstruction we determined that somewhat corrupted data merely introduced a different set of singular values to reconstruct. Furthermore, as the Reuters and ML1M datasets were intact, we used them as controls against the corruption of the other sets.

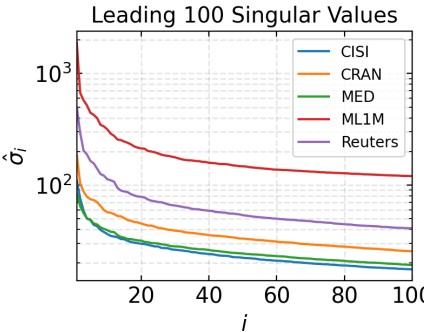

| Dataset | Rows | Columns | nnz(A)/row |
|---|---|---|---|
| CISI [1] | 5609 | 1460 | 12.17 |
| CRAN [1] | 4612 | 1398 | 18.06 |
| MED [1] | 5831 | 1033 | 8.92 |
| ML1M [8] | 6040 | 3952 | 165.60 |
| Reuters-21578 [2–5] | 18933 | 8293 | 20.57 |

Figure 1: Leading 100 singular values for each dataset.

Table 1: Number of rows, columns, and average non-zero elements in each row for datasets.

---

[3]https://github.com/edoliberty/frequent-directions

### 4.3 Experiments

We conducted two sets of experiments: one to confirm the results of [11] in a series of reproducibility studies and another to further measure the performance of the algorithms using two additional metrics as well as observing the effect of the number of batches on the runtime and performance.

**Update method comparison**  As a first step, we sought to reproduce the results in Figures 3 and 4 of [11]. To do this, we conducted the sequence updates experiment. The initial matrix $B \equiv A^{(0)}$ was set equal to the first $\mu$ rows of $A \in \mathbb{C}^{m \times n}$ and the remaining $m - \mu$ rows of $A$ were appended to the initial matrix over a sequence of $\phi$ updates, each with $\tau = \lfloor (m - \mu)/\phi \rfloor$ rows. Following the notation of [11], the $i$-th update would yield $A^{(i)} = \begin{pmatrix} B \equiv A^{(i-1)} \\ E \equiv A(\mu + (i - 1)\tau + 1 : \mu + i\tau, :) \end{pmatrix}$ with the exception of the last update which is likely to have fewer rows in $E$. After each update, the rank-$k$ truncated SVD was calculated by one of the three algorithms.

The parameters used in [11], and thus in our experiments as well were $\mu = \lceil m/10 \rceil$ rows, $\phi = 10$ updates, and rank $k = 50$. The relative errors and residual norms were reported for the $k = 50$ leading singular triplets for $\phi = 1, 5, 10$. For Algorithm 2.2, we set the coefficient $\lambda = 1.01\widehat{\sigma}_1^2$ and $r = 10$.

**Algorithm 2.2 $r$ parameter study**  Next, we varied the $r$ parameter in Algorithm 2.2 to evaluate its effect on the accuracy as was presented in Table 4 by [11]. For this, we set $\mu = \lceil m/10 \rceil$, $\phi = 10$, and $k = 50$ for all three update methods as with the previous experiment and set $r = 10, 20, 30, 40, 50$ for Algorithm 2.2.

**Runtime comparison**  We compared the runtimes of the algorithms for the CRAN, CISI, and MED as a function of the rank $k = 25, 25, 50, 75, 100, 125$ and the total number of updates $\phi = 2, 4, 6, 8, 10$ (Figure 2 left and middle plots in [11]).

**Varying number of batches and desired rank**  In addition to the experiments that we replicated based on [11], we also varied the number of batches $\phi = 2, 4, 6, 8, 10$ and the desired rank $k = 25, 50, 75, 100, 125$ of the truncated SVD and evaluated the performance of each of the update methods to further observe the effects of each of these parameters on the methods' performances.

## 5 Results

**Relative error and residual norms of singular triplets**  The relative error and residual norm of the leading $k = 50$ singular triplets for the CRAN dataset at $\phi = 1, 5, 10$ using Algorithms 2.1, 2.2, and `FrequentDirections` are shown in Figure 2. Due to the large number of figures, the complete set of plots for the standard experiments are presented in Sections A to E in the Supplementary Materials. When comparing the relative error and residual norm plots for Algorithm 2.1 on CRAN, CISI, and MED, our results matched those of [11] exactly. For Algorithm 2.2, the plots did not match exactly, though the differences never exceeded half an order of magnitude and are attributable to the randomness inherent in Algorithm 2.2.

Our comparison of the relative error and residual norm of the $k = 50$-th singular triplet for Algorithm 2.2 with various values of $r$ revealed a similar result to [11] – across the three methods, Algorithm 2.2 had the lowest errors, and within variations of Algorithm 2.2, larger values of $r$ yielded higher accuracy.

**Runtime**  For all three of the datasets which we measured runtimes on, we found Algorithm 2.2 to require a substantially longer amount of time to complete all of its updates. Algorithm 2.1 and `FrequentDirections` required a similar length of time, though Algorithm 2.1 was consistently faster than `FrequentDirections` by a small margin. The runtime plots for the standard experiments are shown in Section F of the Supplementary Materials.

**Number of batches and rank**  Due to space-related constraints, we chose to only include two examples from the array of plots generated (Figure 4). Despite the large variation in the parameters, we can see that the residual norm for overlapping update numbers and $k$ share very similar values.

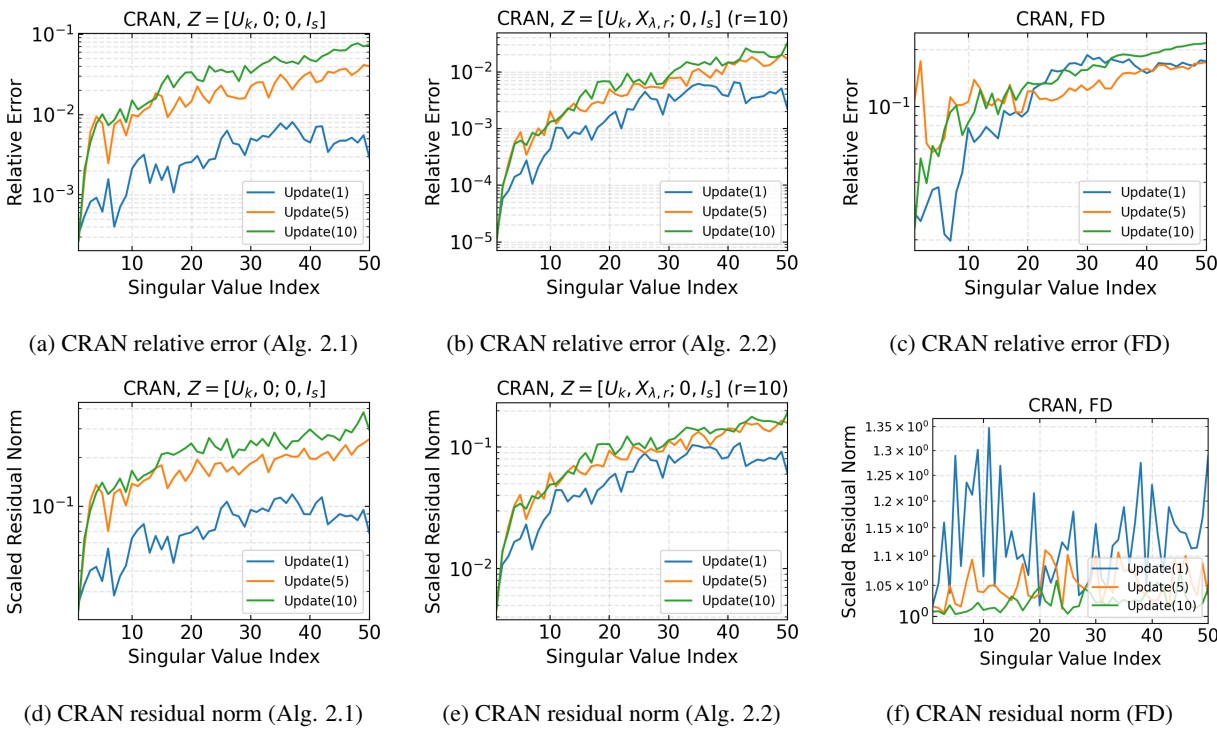

(a) CRAN relative error (Alg. 2.1)  (b) CRAN relative error (Alg. 2.2)  (c) CRAN relative error (FD)

(d) CRAN residual norm (Alg. 2.1)  (e) CRAN residual norm (Alg. 2.2)  (f) CRAN residual norm (FD)

Figure 2: Relative errors and residual norms at $\phi = 1, 5, 10$ for CRAN with Algorithm 2.1, Algorithm 2.2, and FD.

|  | $r$ | MED err. | MED res. | CRAN err. | CRAN res. | CISI err. | CISI res. |
|---|---|---|---|---|---|---|---|
| | 10 | 0.037 | 0.204 | 0.031 | 0.174 | 0.038 | 0.224 |
| | 20 | 0.028 | 0.172 | 0.021 | 0.144 | 0.019 | 0.149 |
| $Z = \begin{pmatrix} U_k & X_{\lambda,r} \\ & & I_s \end{pmatrix}$ | 30 | 0.021 | 0.154 | 0.012 | 0.113 | 0.014 | 0.119 |
| | 40 | 0.015 | 0.133 | 0.010 | 0.107 | 0.011 | 0.105 |
| | 50 | 0.013 | 0.121 | 0.008 | 0.097 | 0.009 | 0.096 |
| $Z = \begin{pmatrix} U_k & \\ & I_s \end{pmatrix}$ | – | 0.101 | 0.294 | 0.074 | 0.295 | 0.080 | 0.382 |
| FrequentDirections | – | 0.212 | 1.031 | 0.216 | 1.045 | 0.205 | 1.032 |

Table 2: Relative error and residual norm of approximation of the singular triplet $\left(\widehat{u}^{(50)}, \widehat{v}^{(50)}, \widehat{\sigma}_{50}\right)$

## 6   Discussion

Ultimately, the reproduced results confirm the original results. Specifically, Table 2 verifies that Algorithm 2.2 outperforms Algorithm 2.1 in terms of accuracy. Furthermore, Figure 3 clearly demonstrates that Algorithm 2.1 far outperforms Algorithm 2.2 with respects to wall clock speed. However, as there were no benchmarks, we viewed the comparison with `FrequentDirections` as a much stronger barometer. At first glance, Table 2 and Figures 2c and 2f suggest that both Algorithm 2.1 and 2.2 outperform `FrequentDirections` in terms of accuracy. However, upon considering the steps involved in `FrequentDirections` (namely the step involving the thresholding of the singular values), we realize that the relative error and residual norm of singular triplets may not be an applicable metric for `FrequentDirections`. This is further demonstrated by the irregular profile of the residual norm as a function of the singular value index (Figure 2f)). Thus it cannot conclusively be said that `FrequentDirections` is significantly under-performing the paper's proposed algorithms. Consequently, the overall conclusion becomes that while the results

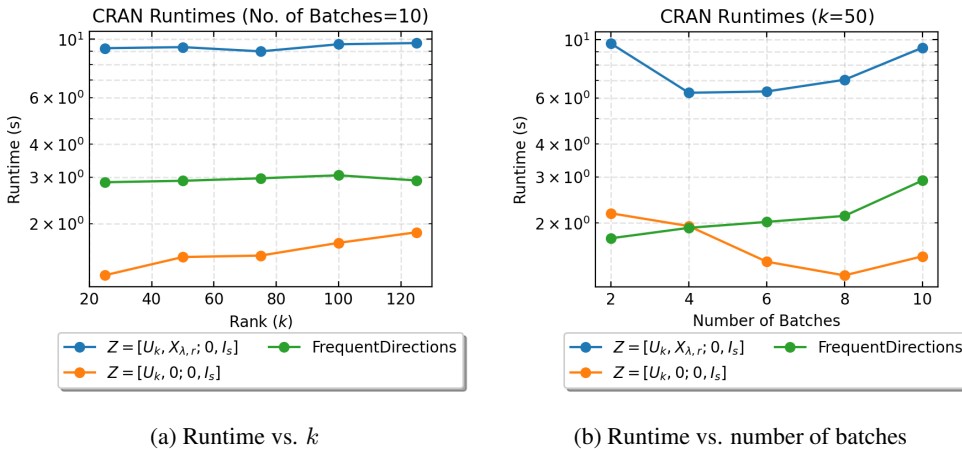

(a) Runtime vs. $k$

(b) Runtime vs. number of batches

Figure 3: CRAN runtimes as a function of rank $k$ (left) and number of batch splits (right).

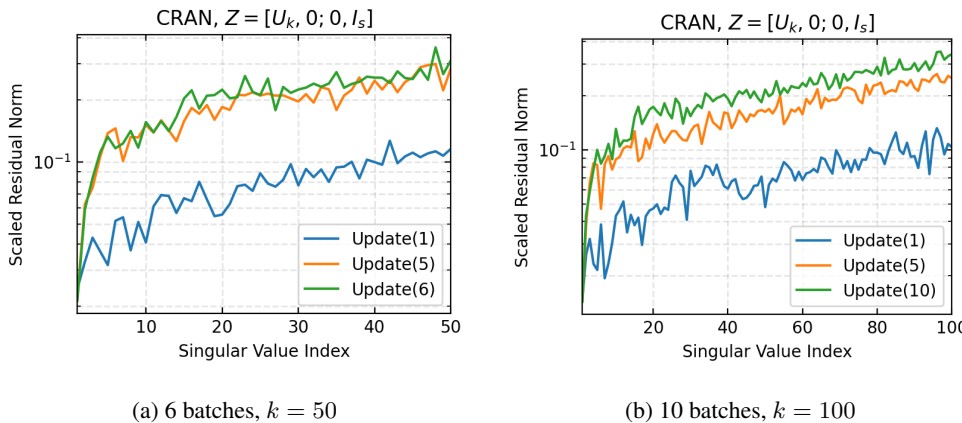

(a) 6 batches, $k = 50$

(b) 10 batches, $k = 100$

Figure 4: Examples of residual norm for experimental parameters outside of what was investigated by [11].

presented in the paper are sound, there is still need for further benchmarking to determine where the proposed algorithms stand relative to the state-of-the-art in the field.

## 6.1 Future Work

We believe a weakness of the paper to be the lack of benchmarking - and as discussed above, our results do not conclusively resolve this. However, they do motivate the need for metrics that will allow for a fair comparison between the proposed algorithm and state-of-the-art algorithms such as `FrequentDirections`.

## 6.2 What was easy

Algorithm 1.1 was quite simple to understand and implement, and was exactly reproduced quite early on. Once we received code, implementation of Algorithm 2.2 and the evaluation metrics was simplified.

## 6.3 What was difficult

In addition to the challenges constructing $X_{\lambda,r}$ for Algorithm 2.2, another challenging/time-consuming aspect was designing the experiments as sweeping through various combinations of the parameters required thorough planning for data management.

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
