# OpenReview forum: "[Re] Projection-based Algorithm for Updating the TruncatedSVD of Evolving Matrices"
_ML_Reproducibility_Challenge/2021/Fall — RC2021_

### Official Review · Reviewer_6mab · 2022-02-25
**Good study, a little bit over detailed**

**Rating:** 8
**Confidence:** 4

**Review:**

The authors did a good job reproducing the algorithm. It is a simple algorithm, so it wasn't very difficult to code it. I appreciate that they took the time to run extensive experiments. The plots justify their claims. In short they reproduce the performance of the new algorithm, but they cannot show that it overperforms the state of the art, as described in the paper
I think the authors spent too much space re-explaining the algorithms. They could have just put a reference. Other than that I think it is a good reproducibility report.

---

### Official Review · Reviewer_CP2G · 2022-03-19
**Projection-based Algorithm for Updating the TruncatedSVD of Evolving Matrices**

**Rating:** 8
**Confidence:** 4

**Review:**

The author wants to verify the central claim and runtime performance of the original paper. The author's analysis of the zha-simon algorithm is correct but it is limited to certain conditions only. The author needs to check one more time. The author would have specified the conditions where the QR and SVD are expensive. The author would have specified the cases where their algorithm is more accurate.  The substantial statements or proofs are required to eliminate QR decomposition in the proposed algorithm. The results are good.

---

### Meta-Review · Program_Chairs · 2022-04-09

**Recommendation:** Accept
**Confidence:** 5

**Metareview:**

The authors have provided a solid contribution.  It has been accepted.

---

### Decision · Program_Chairs · 2022-04-09

**Decision:**

Accept

**Comment:**

Following the recommendation of reviewers and meta-reviewer, the paper is accepted for ML Reproducibility Challenge 2021, and will be published in the upcoming special edition of ReScience Journal.